# Malnutrition in Patients with Liver Cirrhosis

**DOI:** 10.3390/nu13020540

**Published:** 2021-02-07

**Authors:** Julia Traub, Lisa Reiss, Benard Aliwa, Vanessa Stadlbauer

**Affiliations:** 1Department of Clinical Medical Nutrition, University Hospital Graz, 8036 Graz, Austria; Julia.Traub@klinikum-graz.at (J.T.); lisamaria.reiss@edu.fh-joanneum.at (L.R.); 2Department of Gastroenterology and Hepatology, Medical University of Graz, 8036 Graz, Austria; benard.aliwa@medunigraz.at

**Keywords:** malnutrition, cirrhosis, nutritional screening, nutritional assessment, gut–liver axis, macronutrients, micronutrients, dysbiosis

## Abstract

Liver cirrhosis is an increasing public health threat worldwide. Malnutrition is a serious complication of cirrhosis and is associated with worse outcomes. With this review, we aim to describe the prevalence of malnutrition, pathophysiological mechanisms, diagnostic tools and therapeutic targets to treat malnutrition. Malnutrition is frequently underdiagnosed and occurs—depending on the screening methods used and patient populations studied—in 5–92% of patients. Decreased energy and protein intake, inflammation, malabsorption, altered nutrient metabolism, hypermetabolism, hormonal disturbances and gut microbiome dysbiosis can contribute to malnutrition. The stepwise diagnostic approach includes a rapid prescreen, the use of a specific screening tool, such as the Royal Free Hospital Nutritional Prioritizing Tool and a nutritional assessment by dieticians. General dietary measures—especially the timing of meals—oral nutritional supplements, micronutrient supplementation and the role of amino acids are discussed. In summary malnutrition in cirrhosis is common and needs more attention by health care professionals involved in the care of patients with cirrhosis. Screening and assessment for malnutrition should be carried out regularly in cirrhotic patients, ideally by a multidisciplinary team. Further research is needed to better clarify pathogenic mechanisms such as the role of the gut-liver-axis and to develop targeted therapeutic strategies.

## 1. Introduction

The Hepahealth report from 2018 reported a prevalence of chronic liver disease and cirrhosis in Europe between 500 and 1100 cases per 100.000 inhabitants [1]. Data from the USA show a 65% increase in cirrhosis associated mortality between 1999 and 2016 [2]. Cirrhosis is a systemic disease and malnutrition is a key feature as well as an important complication of the disease. This implicates that malnutrition diagnosis is not only relevant as one of the clinical characteristics of cirrhosis, but also needs to be considered as an important complication, that warrants timely and appropriate therapy to improve prognosis. Our review should highlight the importance of early diagnosis, should help to understand the pathophysiology and define appropriate therapeutic measures. Additionally, knowledge gaps are identified. In this review the term malnutrition is used to describe undernutrition. The discussion of overnutrition in liver disease is beyond the scope of this review.

The reported prevalence of malnutrition in cirrhosis is highly variable, ranging from 5–92%, indicating either a knowledge gap or difficulties in diagnosing malnutrition or both. The knowledge gap is underpinned by a survey, in which only 20% of gastroenterologists gave correct answers regarding the prevalence of malnutrition in cirrhotic patients [3]. The problem of underdiagnosis can be derived from two very large studies from the USA using the national inpatient sample, that showed a much lower prevalence of 6–12%, whereas studies that used active screening for malnutrition exhibit higher rates of detection [4,5]. Appendix A summarizes studies describing the prevalence of malnutrition in cirrhosis over the past five years, depicting the large differences depending on the study design and the tools used for diagnosis. Malnutrition should therefore be routinely screened for and assessed among this group of high-risk patients to avoid underdiagnosis. However, screening of malnutrition in liver cirrhosis patients is challenging because of the influence of fluid retention, ascites and peripheral edema [5]. Malnutrition prevalence, as a key feature of cirrhosis, increases with increasing disease severity [6,7]. However, frequently also patients with compensated liver disease are malnourished and malnutrition is a considerable risk factor for mortality even in patients with a model of end-stage liver disease (MELD) score < 15 [8]. Furthermore, patients with chronic liver disease but without cirrhosis are also frequently malnourished [9]. In this population, malnutrition may often be masked by obesity [9]. Malnutrition can also be seen as a complication of cirrhosis, since it has a negative impact on disease progression and outcome. Rate of hospitalization and mortality is doubled in malnourished compared to adequately nourished patients and malnutrition is an independent predictor of outcome [10,11,12]. Malnutrition is a predictor of other complications of cirrhosis [13]. Especially infections and hepatic encephalopathy are associated with malnutrition [14,15]. Furthermore, other malnutrition related diagnoses, namely sarcopenia, hepatic osteodystrophy and frailty are commonly found in liver cirrhosis. The presence of sarcopenia further impairs prognosis [16,17,18,19,20,21,22,23,24,25]. The interplay between malnutrition, sarcopenia and frailty has recently been reviewed [26].

This review summarizes the current knowledge on pathogenesis of malnutrition in cirrhosis and discusses the best clinically applicable strategies to diagnose malnutrition in order to raise awareness for this still often underappreciated complication of cirrhosis. Treating malnutrition by a multidisciplinary team improves survival rates and quality of life in patients with liver cirrhosis [27] and nutritional education leads to more nutrition consultations and a lower 90-day readmission rate in cirrhosis [28]. Therefore this review also discusses current and potential future therapeutic options to treat malnutrition in cirrhosis.

## 2. Pathogenesis

The etiology of malnutrition in liver cirrhosis is multifactorial (Figure 1). Decreased energy and protein intake, inflammation, malabsorption, altered nutrient metabolism, hormonal disturbances, hypermetabolism and gut microbiome dysbiosis can contribute to malnutrition. Additionally, fasting periods and external factors such as alcohol consumption have impact on malnutrition.

### 2.1. Decreased Energy and Protein Intake

In patients with liver cirrhosis, decreased energy and protein intake are the commonest reason leading to malnutrition [29,30,31,32,33,34,35]. The percentage of patients with inadequate energy intake ranges from 9.2% to 100% in different studies, depending on the method of assessment and the patient population. Energy intake is reduced by 13–34%, indication a large variation in different studies. Appendix A shows the design and results of different studies assessing nutritional intake. Several upstream mechanisms are known as reasons for decreased energy and protein intake (Figure 2). Impaired gastric motility and relaxation due to portal hypertension leads to reduced nutritional intake [36,37]. The presence of ascites can reduce food intake due to early feeling of fullness [38]. A decreased sense of smell and/or dysgeusia, which can be caused by micronutrient deficiencies can also be responsible for a decreased intake [39,40]. Additionally, recommended dietary restriction like a low-salt diet are discussed as possible factors for inadequate nutritional intake [41,42]. In cirrhosis, interestingly, high levels of ghrelin were observed [31,43,44]. Ghrelin is the only known peripherally-derived orexigenic hormone that normally increases appetite and food intake. However, despite high ghrelin levels, appetite is not increased in cirrhotic patients. High ghrelin levels can therefore be considered as an ineffective compensatory mechanism in cirrhosis [31]. To date, it is unknown, which of these factors plays the most important role. Therefore, all factors should be considered during the assessment of nutritional intake in each patient as a personalized approach to detect and adequately treat the most likely reasons for reduced nutritional intake.

### 2.2. Malabsorption and Altered Metabolism of Macro- and Micronutrients

Fat malabsorption is commonly seen in cirrhosis [45,46]. Impaired bile acid metabolism which affects the formation of micelles that are necessary for fat digestion and absorption of fat-soluble vitamins, [47,48] small intestinal bacterial overgrowth, which is common in liver cirrhosis patients, [49] can lead to fat malabsorption via deconjugation of bile acids [50]. Chronic pancreatitis, secondary to alcohol abuse and common in liver cirrhosis patients, may contributes to fat malabsorption as well [51,52]. Protein loss due to portal hypertensive enteropathy has been described [53,54,55]. No data is available regarding impairment of carbohydrate absorption in cirrhosis. Malabsorption needs to be considered in the nutritional assessment and diagnosed, using biomarkers such as fecal elastase or fecal alpha-1-antitrypsin and tests for micronutrient deficiencies (see below). A useful stepwise diagnostic algorithm, starting with noninvasive routine blood tests and specific biomarkers has been proposed by Nikaki K. in 2016 [56].

In addition to altered absorption, also fat, protein and carbohydrate metabolism are altered in cirrhosis, with differing mechanisms depending on the etiology. Much research has been done to elucidate the mechanisms: chronic alcohol consumption alters lipid metabolism by stimulating lipogenesis, decreasing the export of very low-density lipoprotein, activate de novo lipogenesis and inhibiting fatty acid oxidation which also contributes to alcoholic fatty liver disease [57,58,59,60,61]. Alcohol consumption also impairs fatty acid catabolism predominantly through inhibition of mitochondrial ß-oxidation, which is the most significant contribution to alcohol-induced hepatic lipid accumulation and leads to triglyceride accumulation in the liver [62,63,64]. Also in non-alcoholic fatty liver disease, adipose tissue and hepatic triglyceride metabolism is altered [65]. Protein metabolism is impaired due to increased protein catabolism and decreased protein synthesis. Furthermore, decreased serum branched chained amino acids (BCAA) concentration and increased levels of aromatic amino acids are observed in cirrhosis, which play a role in the pathogenesis of hepatic encephalopathy and muscle wasting [66,67,68,69,70]. Glucose metabolism is severely altered as well: peripheral insulin resistance but normal or enhanced uptake into the liver as well as alterations in glycolytic enzymes and changes in glucose and insulin transporters have been described. This contributes to decreased hepatic glucose production and lower hepatic glycogen reserves, associated with increased gluconeogenesis from amino acids and secondary protein breakdown. The above described metabolic abnormalities in carbohydrate metabolism also lead to a state of accelerated starvation already after an overnight fast [71,72,73,74,75,76]. The role of portal hypertension and portosystemic shunting in protein energy metabolism is not fully elucidated yet: on the one hand, the placement of a transjugular intrahepatic portosystemic shunt can lead to improvement of fat free mass and thereby prognosis [77,78,79]. On the other hand, there is also evidence that portosystemic shunting may have deleterious nutritional effects due to a reduction in hepatic nutrient flow [80]. While the molecular principles of changes in macronutrient metabolism in cirrhosis are already well described, the direct therapeutic implications of these findings are not well defined yet. Timing and composition of meals as therapeutic measures to account for changes in macronutrient metabolism are described below. Further research is needed to understand the effect of portal hypertension and portosystemic shunting in humans on protein anabolism and catabolism including the role of the gut-liver axis.

But not only macronutrient metabolism is altered in liver cirrhosis patients. Deficiencies in trace elements, minerals and vitamins are common, due to fat malabsorption, diuretic use and inadequate intake. In addition, liver dysfunction itself can lead to alterations in trace element metabolism [81,82]. Zinc, selenium, iron and magnesium are commonly decreased in liver cirrhosis [83,84,85,86,87,88] whereas copper and manganese can be increased [81,89]. Fat-soluble vitamin deficiencies are common in liver cirrhosis [90] which can in turn impair absorption of other nutrients, such as protein and fat [91]. For the absorption of fat-soluble vitamins, bile acids are required to form micelles which are absorbed by enterocytes into the circulation. If there is inadequate delivery of bile acids, as it is common in liver cirrhosis patients, this can lead to a deficiency of fat-soluble vitamins, especially in jaundiced patients [92,93]. Trace element and vitamin deficiencies can in turn impact negatively on nutrition intake indicating a vicious cycle of malnutrition in cirrhosis: zinc and vitamin A deficiency can impair taste and olfaction and therefore impair food intake. [39,94] Vitamin D deficiency is of prognostic relevance, since it is associated with poor outcome, increased mortality and higher complications rate, however it is yet unclear whether vitamin D levels are a mere surrogate of advanced liver disease or if there exists a direct pathophysiological relation [95,96,97,98]. Water-soluble vitamins, especially vitamins C, B1, B2, B6 and folic acid [99,100,101] are decreased whereas vitamin B12 levels can be falsely increased in liver cirrhosis, possibly due to a flooding of vitamin B12 from damaged liver cells into the circulation [102,103]. For a summary of changes of micronutrients in cirrhosis see Appendix A.

Not only are absorption and metabolism altered, but also energy expenditure contributes to malnutrition: 15–30% of cirrhotic patients are hypermetabolic with a resting energy expenditure of >120%, which negatively effects nutrition status [104,105,106,107]. Hypermetabolism compromises overall transplant-free and early post-transplant survival [108,109,110]. The cause of hypermetabolism in liver cirrhosis is not yet clarified in full detail. From rheumatoid disease it is known that inflammation drives hypermetabolism [111]. Elevated levels of interleukin-1, interleukin-6 and transforming growth factor are also common in chronic alcoholic liver diseases [112,113,114,115,116]. Therefore, inflammation can be considered as a contributing factor to hypermetabolism and malnutrition [117], alongside with increased beta-adrenergic activity [72]. Additionally, elevated levels of proinflammatory cytokines may be directly responsible for decreased appetite [118,119]. Since inflammation in cirrhosis is tightly linked to changes in the gut–liver axis, [120,121,122] the relation to hypermetabolism needs further research to elucidate pathophysiology and define possible therapeutic interventions.

Hormones, as superordinate control of nutritional intake and metabolism also impact on nutrition in cirrhosis. The role of ghrelin in appetite regulation was already described above. In addition, ghrelin has a wide spectrum of other metabolic functions in glucose metabolism and weight control and posttranslational modification is essential to exert its metabolic function. The diverse roles of ghrelin in liver disease has recently been extensively reviewed [123]. Ghrelin as well as leptin are known to influence energy expenditure [124,125]. Leptin, which helps to regulate energy balance, circulates in free and bound form. The basal concentrations of leptin are higher in patients with liver cirrhosis and can lead to inadequate energy expenditure [126]. Hyperinsulinemia and insulin resistance are also common in liver cirrhosis; increased insulin levels induce satiety, leading to a reduction in energy intake [127]. Testosterone is reduced in about 90% of men with liver cirrhosis [128] and plays an important role in protein synthesis and protein breakdown [129].

### 2.3. Gut Microbiome Dysbiosis as Potential Contributor to Malnutrition

Altered nutritional status is associated with distinct gut microbiome dysbiosis in cirrhosis [130]. The gut microbiome is a nutrient signal transducer with the capacity to synthesize or modify nutrient signaling molecules such as short-chain fatty acid (SCFA) and branched chain amino acid (BCAA) [131,132]. Several bacterial genera are known to produce SCFA, such as *Bacteroides*, *Faecalibacterium*, *Succinivibrio* and *Butyricimonas* among others [133,134]. Undernourished children for example showed a lower abundance of different *Bacteroides* species, suggesting a loss in SCFA-producing species [135,136]. In cirrhosis, SCFA-producing bacterial species are reduced [137]. The observed alteration in the gut microbiome composition in cirrhosis is associated with increased protein catabolism mediated by inflammatory responses leading to muscle loss [138]. Gut microbiome dysbiosis is further associated with increased gut permeability and bacterial translocation, which is associated with inflammation [122] and complications of cirrhosis [122,139,140]. It is not known to date whether gut microbiome dysbiosis precedes the development of malnutrition in cirrhosis or if it is a consequence of the disease and the drug treatment of the disease. This question would be of high relevance to answer, in order to develop microbiome targeted therapeutic strategies to improve malnutrition in the clinical setting.

## 3. Diagnosis

Since malnutrition is a common key feature and a complication of liver cirrhosis and related to a poor prognosis, early diagnosis is important. Unfortunately, the common models to determine the prognosis in patients with liver cirrhosis, such as the model of end-stage liver disease (MELD) [141] and the Child–Pugh score [142], do not include nutritional screening or assessment. Of note, the original score developed by the surgeons Child and Turcotte, contained malnutrition as a variable, which was later substituted by prothrombin time [143]. All liver cirrhosis patients should be rapidly prescreened for the risk of malnutrition at each contact by assessing Child–Pugh score and BMI and when at high risk (Child–Pugh score C irrespective of the BMI or BMI < 18.5 kg/m^2^ irrespective of the Child–Pugh score) a nutritional assessment, including assessment of sarcopenia as a complication of malnutrition, should be completed immediately to confirm the presence and determine the severity of malnutrition [144,145,146]. This prescreening can be done by skilled personnel from different disciplines, since it contains routine clinical data, which are normally collected at each outpatient visit or at hospitalization. In the future, automated prescreening combining routinely assessed data from electronic patient records, is thinkable. A specific screening tool, which is validated for patients with liver cirrhosis, is advised to account for special circumstances such as fluid retention in cirrhosis. The Royal Free Hospital–Nutritional Prioritizing Tool (RFH–NPT) fulfills this requirement [145,147]. Common malnutrition screening tools for the general hospital population do not perform well in cirrhosis [148]. If a general malnutrition tool is intended to be used in cirrhosis, it needs to be validated first. In patients with medium or high risk in the RFH–NPT (1 point or more), a detailed nutritional assessment using cirrhosis specific assessment tools such as the Subjective Global Assessment or the Royal Free Hospital Global Assessment as well as a detailed assessment of dietary intake are required [145,147]. In patients, who are at high risk for malnutrition, the assessment of sarcopenia in addition to a detailed nutritional assessment is recommended to confirm and characterize complications of malnutrition and identify modifiable variables for nutrition support. Methods for the assessment of sarcopenia in liver cirrhosis have recently been reviewed [149]. In obese patients with cirrhosis Child–Pugh A or B (BMI > 30 kg/m^2^), a nutritional and lifestyle intervention targeting obesity is indicated [145,147]. As nutritional assessment is more comprehensive, time consuming and requires interpretation of multiple nutrition indicators, it should be performed by a dietitian [150]. In patients with low risk of malnutrition, rescreening should be performed every year, in all other patients the assessment should be repeated every one to six months in the outpatient setting and at admission and for inpatients periodically during the hospital stay [145]. Figure 3 shows a comprehensive algorithm to screen for and assess malnutrition in cirrhosis, adapted from the European Association for the Study of the Liver (EASL) clinical practices guidelines [145].

## 4. Therapeutic Strategies

### 4.1. Diet

The positive effect of dietary interventions on prognosis in cirrhosis has been clearly shown: optimizing the nutritional status of liver cirrhosis patients improves morbidity and mortality [13,17,151,152,153] even in patients with acute on chronic liver failure [154]. Nutritional therapeutic interventions by a multidisciplinary team, especially through dietary counseling from dieticians, improves biomarkers of malnutrition, [155,156] quality of life [27,157] and survival rate [27].

The recommended macronutrient composition in cirrhosis mainly focusses on protein intake. Cirrhotic patients have an increased protein requirement based on the increased protein turnover and catabolism [158,159]. A high protein intake improves in nutritional status [155,160]. Even patients with hepatic encephalopathy, where in the past protein restriction was advocated, [161] benefit from normal to high protein intake [157,162,163,164,165]. The recommendations regarding protein intake differ slightly, but not relevantly, between different guidelines, depending on the nutritional status and range from 1.2–1.5 g protein/kg bodyweight (Table 1). There are no specific recommendations regarding carbohydrate and fat intake for patients with liver cirrhosis.

Current evidence suggests that timing and frequency of meals is of importance to improve malnutrition in cirrhosis. After overnight fasting, glycogen stores in cirrhotic livers are emptied [74]. A late evening snack with 50 g complex carbohydrates can improve nitrogen metabolism, increase lean body mass and reverse anabolic resistance and sarcopenia [166,167,168,169]. Eating breakfast improves cognitive function in cirrhosis, indicating that, depending on personal preferences in dietary habits, an individualized approach for timing of meals should be developed during dietary counselling [170]. A higher frequency of 5–6 meals/day also shortens episodes of catabolism during the day [171]. It is particularly important to pay attention to the timing and frequency of meals in hospitalized patients to avoid long and often unnecessary “nil per os” periods due to planned diagnostic tests.

The main emphasis in dietary counselling should be to ensure adequate oral intake. If oral intake including oral nutrition supplements (see below) is insufficient despite adequate nutritional advice, enteral tube feeding may be considered for cirrhotic patients to achieve their nutritional and energy goals [172]. Nonbleeding esophageal varices are no contraindication for the placement of a nasogastric tube [163,173]. An endoscopic gastrostomy, on the other hand, is associated with a higher risk of complications, especially bleeding, and is therefore not recommended for patients with advanced chronic liver disease [174]. In moderately or severely malnourished cirrhosis patients, who are unable to eat oral food or cannot be fed sufficiently enterally, parenteral nutrition should be started according to the ESPEN recommendations [175]. Additionally, parenteral nutrition should be given when fasting periods last longer than 72 h [175]. Since cirrhotic patients are more prone to sepsis or infections, care should be taken to avoid infections from central venous lines [176].

When ascites is diagnosed in patient with liver cirrhosis, a “no added salt” diet restricted to 90 mmol salt per day (5.2 g) is recommended by many guidelines. However, as we recently reviewed in detail, a low-sodium diet, although leading to a faster disappearance of ascites and less need for diuretics can lead to poor diet adherence because of impaired taste of the meals, reduced energy and protein intake and increased risk of malnutrition [42]. Therefore the risks and benefits of salt restriction have to be weighed carefully in each patient, again showing the necessity of a multidisciplinary team approach to treat malnutrition in cirrhosis.

### 4.2. Oral Nutritional Supplements and Micronutrients Supplementation

Oral nutritional supplements can help to achieve nutritional goals in cirrhosis. A significant improvement in anthropometric nutritional parameters such as lean muscle mass and body mass index as well as serum proteins can be achieved by oral nutritional supplements [160,177]. A meta-analysis concluded that oral nutritional supplements may also improve outcome [152]. Furthermore, an improvement in quality of life, functional status and rehabilitation of malnourished cirrhosis patients can be achieved [157,178]. In terms of administration time, nocturnal oral nutritional supplements have a better effect in improving the total body protein status than at daytime [166] (Table 2). Since oral nutritional supplements also contain micronutrients and vitamins, this may be of additional benefit in patients with cirrhosis, however, so far no clear benefit of micronutrient supplementation could be shown [179]. Moreover, no clinical data comparing different products are available, therefore choice can be based on personal preference and price.

Although some studies demonstrate the positive effect of micronutrients supplementation, due to the lack of robust data, no clear recommendations can be made. Additional supplementation is currently only recommended in cases of confirmed or clinically suspected deficiency. Data on zinc supplementation is conflicting: some studies report positive effects on zinc supplementation in hepatic encephalopathy [180,181,182], while others report no significant improvements [183,184]. Zinc supplementation in case of deficiency improves liver function and nutritional status [185,186] and may even impact positively on clinical outcome [187,188]. Normalizing zinc and vitamin A levels can also indirectly improve the nutritional status by a positive effect on sense of taste and thereby increased food intake [94,189]. A meta-analysis found no evidence to support or refute antioxidant supplements such as beta-carotene, vitamins A, C, E and selenium in liver disease [179]. Also, for vitamin D supplementation, evidence is not sufficient. Vitamin D deficiency is common in cirrhosis and associated with increased mortality [190] and deficiency can be corrected by oral supplementation in cirrhosis [191,192]. However, due to inadequate overall data quality, there is not sufficient evidence to prove or disprove an effect on morbidity and quality of life of vitamin D supplementation [193] (Table 2).

### 4.3. Amino Acids

BCAA serum levels are low in cirrhosis because they are preferentially used as energy substrates, but are also essential for protein synthesis and ammonia detoxification [194]. BCAA supplementation has been shown to prevent lipolysis and proteolysis and improves nitrogen balance, muscle mass, nutritional status, complication free survival, quality of live and hepatocellular carcinoma risk [194,195,196,197,198,199,200,201,202]. However, despite this quite clear evidence of positive effects, current guidelines recommend BCAA supplementation only in decompensated cirrhosis, when adequate protein intake cannot be achieved by oral diet or in case of complications. For hepatic encephalopathy, vegetarian protein, which is rich in BCAA, is considered to be the ideal protein source, not only because it is better tolerated than animal protein [164,203,204] but also because vegetarian protein may positively influence gut microbiome composition [205]. BCAA can also be used as late evening snack to improve nutritional status [73] especially in protein-intolerant patients [144,145,146,206] (Table 2).

## 5. Summary

Malnutrition is a common and dangerous key feature and complication of liver cirrhosis. Diagnosis is challenging and often overlooked. A repetitive diagnostic workup with a rapid screening for malnutrition using the Child–Pugh score and BMI, and in selected patients the RHF-NPT to stage patients in low, medium and high risk and conducting a detailed nutritional assessment including assessment of complications of malnutrition in patients at medium and high risk should be implemented in every center. Education on malnutrition in cirrhosis for all health care professionals who treat patients with liver cirrhosis and the availability of trained dieticians seem crucial in the future management process of malnutrition in cirrhosis, to account for the complexity of the disease and the need for individualized management. From a pathophysiological point of view, more work is needed to identify the drivers of malnutrition, especially considering the complex interplay between the gut microbiome and nutrient metabolism. Therapeutic efforts should consider alternative pathophysiological mechanisms in the development of malnutrition such as the role of inflammation and dysbiosis to identify potential therapeutic targets beyond the pure increase of nutritional intake. Further clinical studies, ideally as multicenter, multidisciplinary initiatives are needed to diagnose and adequately treat malnutrition in cirrhosis.

## Figures and Tables

**Figure 1 nutrients-13-00540-f001:**
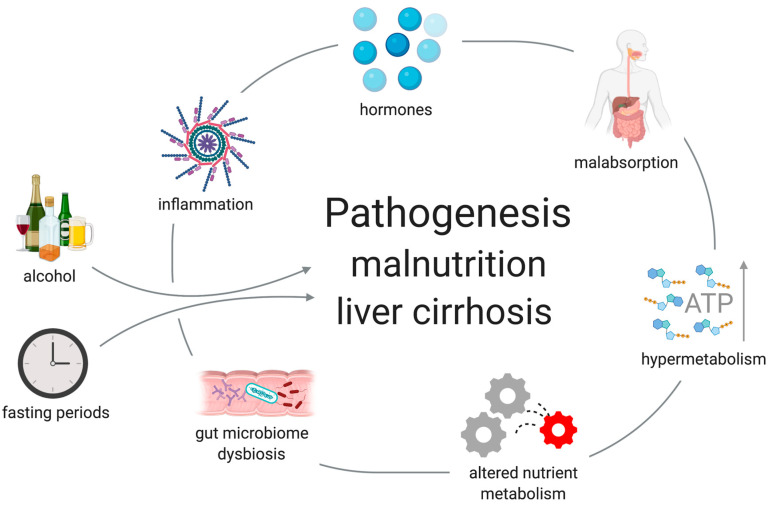
Factors contributing to malnutrition in cirrhosis. Created with BioRender.com.

**Figure 2 nutrients-13-00540-f002:**
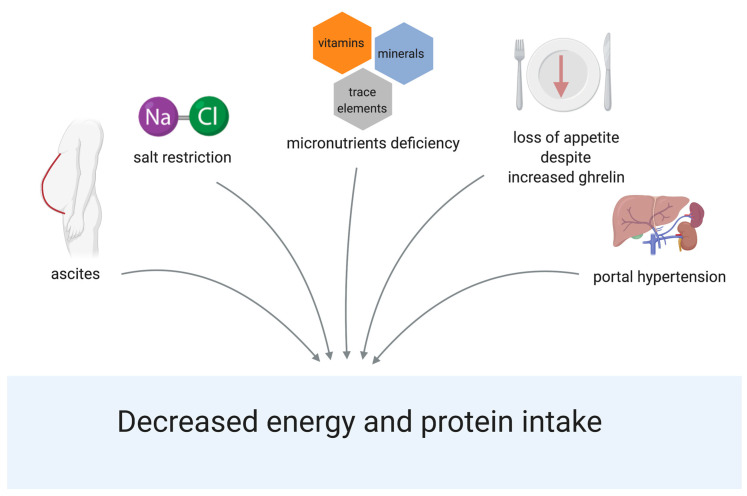
Reasons for decreased energy and protein intake in cirrhosis. Created with BioRender.com.

**Figure 3 nutrients-13-00540-f003:**
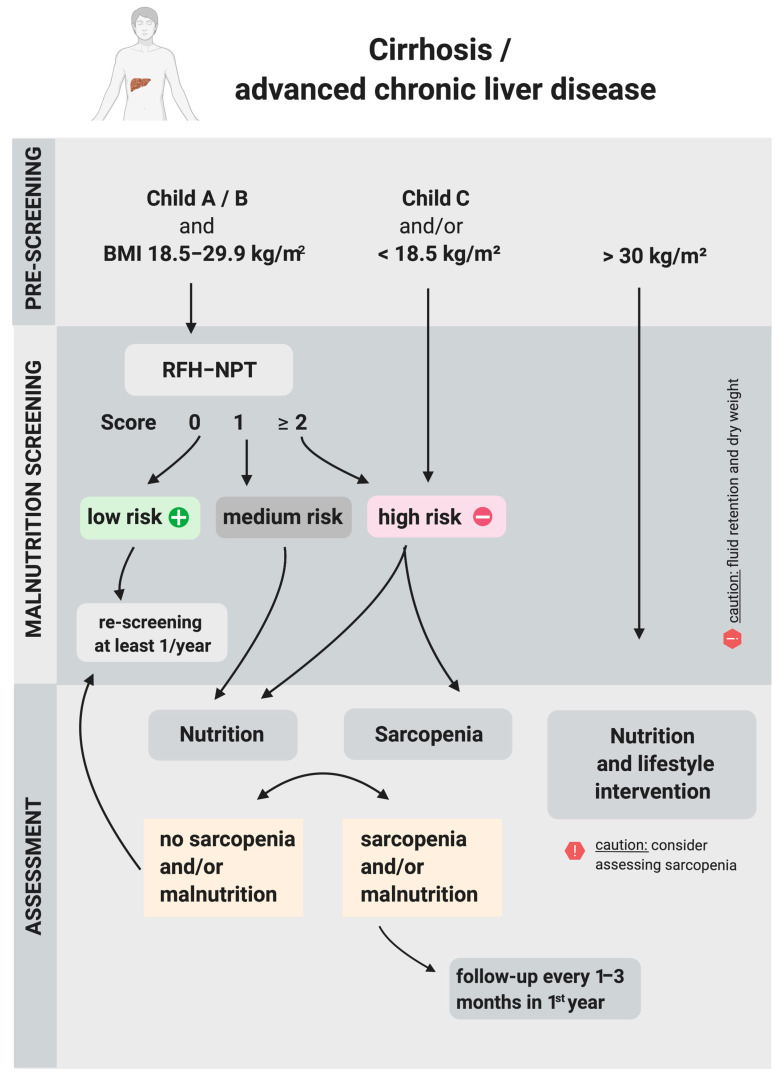
Algorithm to screen for and assess malnutrition in cirrhosis, adapted from the European Association for the Study of the Liver (EASL) clinical practices guidelines [145]. Created with BioRender.com.

**Table 1 nutrients-13-00540-t001:** Summary of dietary recommendations in cirrhosis from different populations.

Energy	General cirrhosis patients	>35 kcal/kg bodyweight ^1^
	Malnourished cirrhosis patients	30–35 kcal/kg bodyweight ^3^
	Cirrhosis patients with HE	35–40 kcal/kg bodyweight ^1,2,4^
Protein	General cirrhosis patients	1.2–1.5 g/kg bodyweight ^1^
	Replenish malnourished and/or sarcopenic cirrhosis patients	1.5 g/kg bodyweight ^3^
	Nonmalnourished cirrhosis patients	1.2 g/kg bodyweight per day ^3^
	Cirrhosis patients with HE	1.2–1.5 g/kg bodyweight ^1,2,4^ no protein restriction ^3^
Fat		No specific recommendations
Carbohydrates		No specific recommendations
	Cirrhosis patients with HE	25–45 g of fiber ^4^ Late evening snack with 50 g complex carbohydrates ^4^
Dietary pattern	General cirrhosis patients	3–5 meals a day ^3^ Late evening snack ^3^
	Malnourished decompensated cirrhotic patients	Late evening ONS and breakfast ^1^
	Cirrhosis patients with HE	Small meals evenly distributed throughout the day ^2,4^ Late- night snack ^2,4^
Salt		“No added salt” diet with 5–6 g salt per day, care should be taken when salt reduction leads to reduced palatability ^1,3^

HE, hepatic encephalopathy; ONS oral nutritional supplements. ^1^ European Association for the Study of the Liver (EASL). ^2^ American Association for the Study of Liver Diseases (AASLD). ^3^ European Society for Clinical Nutrition and Metabolism (ESPEN). ^4^ International Society for Hepatic Encephalopathy and Nitrogen Metabolism Consensus (ISHEN).

**Table 2 nutrients-13-00540-t002:** Summary of recommendations for oral nutritional supplements, micronutrient supplementation and branched chain amino acid (BCAA) supplementation from different populations.

Oral nutritional supplements	General cirrhosis patients	Late evening oral nutritional supplement ^1^
	Cirrhosis patients with HE	Liquid nutritional supplements evenly distributed throughout the day ^2^
Micronutrient supplements	General cirrhosis patients	Administration of micronutrients to treat confirmed or clinically suspected deficiency^1,3^ Assessment of vitamin D levels based on frequent deficiency and supplement at vitamin D levels < 20 ng/mL, to reach serum vitamin D (25(OH)D) > 30 ng/mL ^1^
	Cirrhosis patients with HE	For patients with decompensated cirrhosis or those at risk for malnutrition a 2-week course of a multivitamin preparation could be justified ^4^ Specific treatment of clinically apparent vitamin deficiencies ^4^
Amino acids	General cirrhosis patients	Decompensated cirrhosis: BCAA supplements and leucine enriched amino acid supplements when inadequate nitrogen intake by oral diet ^1^ Oral vegetable proteins or BCAA (0.25 g/kg bodyweight/day) for cirrhosis patients who are protein “intolerant” to facilitate adequate protein intake ^3^ Long-term oral BCAA supplements (0.25 g/kg bodyweight/day) in patients with advanced cirrhosis in order to improve event-free survival or quality of life ^3^
	Cirrhosis patients with HE	BCAA supplementation for improving neuropsychiatric performance and to reach the recommended nitrogen intake ^1^ Oral BCAA supplementation for cirrhosis patients who are “intolerant” of dietary protein to achieve and maintain recommended nitrogen intake ^2,4^

HE, hepatic encephalopathy; BCAA, branched chain amino acids. ^1^ European Association for the Study of the Liver (EASL). ^2^ American Association for the Study of Liver Diseases (AASLD). ^3^ European Society for Clinical Nutrition and Metabolism (ESPEN). ^4^ International Society for Hepatic Encephalopathy and Nitrogen Metabolism Consensus (ISHEN).

## Data Availability

Not applicable.

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
