# Peer review of "Malnutrition in Patients with Liver Cirrhosis"

_nutrients, 2021, doi:10.3390/nu13020540_

Round 1
Reviewer 1 Report
The authors wrote a much-needed well written a comprehensive review on malnutrition in patients with cirrhosis. By the end of the review, I was still felt ill-equipped to diagnose malnutrition. Therefore, I think fleshing out the diagnosis section of the review would be greatly helpful to the clinicians. Besides, it would be great to have the authors elaborate if the term "malnutrition" which includes "undernutrition" and "overnutrition," which the authors term as "obese cirrhosis." At the end of the review, a reader may be interested in hearing from the authors as to what could be done differently- evaluation for malnutrition (initial evaluation?, and then annually?); studies needed to learn more about this entity and treatment.
Further details are listed below:
Abstract:
Line 14: There is a typo – I suspect the authors wanted to state 20-50% of patients.
Introduction:
Line 80: Please separate the sentence describing the percentage of patients in cirrhotic patient cohorts (9.2-100%) and the second half, which described the reduction in energy intake from 13-34%. This may help convey the message more effectively.
Line 89: A word may be missing after normally (increase?) appetite and food intake.
Line 89-90: Again, please consider splitting sentences for easier readability.
Line 98: Word missing after common. "Fat malabsorption is common(ly seen) in cirrhosis."
Line 120: Please correct "brain" to "branched" chain amino acids.
Line 122: The references-101 and 102 do not suggest increased gluconeogenesis in cirrhosis. Instead, reference 104 suggests this finding. I recommend adding the ref 104 to 101 and 102.
Line 127: Ref 104 suggests that the metabolic profile of a patient with cirrhosis who fasted overnight resembles that of someone with a normal liver after much more prolonged starvation. I suggest clarifying the second part of the statement, which appears to be an exaggeration. The second part of the statement contradicts the conclusion by the authors of the original paper – "After an overnight fast hepatic glucose production in patients with cirrhosis was diminished as a result of low-rate glycogenolysis. Hepatic gluconeogenesis and ketogenesis were increased. This pattern of hepatic metabolism mimics that seen in "normal" patients after more advanced stages of starvation. After 3 d of starvation, patients with hepatic cirrhosis have hepatic gluconeogenic and ketogenic profiles comparable to those of normal patients undergoing starvation of similar duration."
Author Response
We thank the reviewer for his/her favorable and valuable comments. The responses to each point can be found below and in the revised manuscript file.
The authors wrote a much-needed well written a comprehensive review on malnutrition in patients with cirrhosis. By the end of the review, I was still felt ill-equipped to diagnose malnutrition. Therefore, I think fleshing out the diagnosis section of the review would be greatly helpful to the clinicians.
Thank you for this suggestion. We rewrote the diagnosis section completely to work out the important steps in screening and assessment more clearly and added a flowchart for practical use. We hope that this fulfills your expectations.
Please see line 207-240
Besides, it would be great to have the authors elaborate if the term "malnutrition" which includes "undernutrition" and "overnutrition," which the authors term as "obese cirrhosis."
Thank you for this important comment. We decided – also in the light of the comments of reviewer 3, not to expand on the topic of overnutrition and we now clarify this in the introduction: “In this review the term malnutrition is used to describe undernutrition. The discussion of overnutrition in liver disease is beyond the scope of this review.”
At the end of the review, a reader may be interested in hearing from the authors as to what could be done differently- evaluation for malnutrition (initial evaluation?, and then annually?); studies needed to learn more about this entity and treatment.
We agree with this suggestion and added the timing of reassessments in the section “diagnosis” and into the flowchart. We further expanded the summary to point out future areas of research.
See lines 235-238 and figure 3
Abstract:
Line 14: There is a typo – I suspect the authors wanted to state 20-50% of patients
Sorry, we corrected the typo
Introduction:
Line 80: Please separate the sentence describing the percentage of patients in cirrhotic patient cohorts (9.2-100%) and the second half, which described the reduction in energy intake from 13-34%. This may help convey the message more effectively.
We separated the sentences as suggested.
Line 89: A word may be missing after normally (increase?) appetite and food intake.
We added “increases”
Line 89-90: Again, please consider splitting sentences for easier readability.
We split the sentences as suggested.
Line 98: Word missing after common. "Fat malabsorption is common(ly seen) in cirrhosis."
Corrected
Line 120: Please correct "brain" to "branched" chain amino acids.
Sorry, the typo was corrected
Line 122: The references-101 and 102 do not suggest increased gluconeogenesis in cirrhosis. Instead, reference 104 suggests this finding. I recommend adding the ref 104 to 101 and 102.
You are right, we corrected the references accordingly and we rewrote this paragraph according to the suggestions of reviewer 3.
Please see line 129-135
Line 127: Ref 104 suggests that the metabolic profile of a patient with cirrhosis who fasted overnight resembles that of someone with a normal liver after much more prolonged starvation. I suggest clarifying the second part of the statement, which appears to be an exaggeration. The second part of the statement contradicts the conclusion by the authors of the original paper – "After an overnight fast hepatic glucose production in patients with cirrhosis was diminished as a result of low-rate glycogenolysis. Hepatic gluconeogenesis and ketogenesis were increased. This pattern of hepatic metabolism mimics that seen in "normal" patients after more advanced stages of starvation. After 3 d of starvation, patients with hepatic cirrhosis have hepatic gluconeogenic and ketogenic profiles comparable to those of normal patients undergoing starvation of similar duration."
We changed the sentence to clarify
Please see line 133-134
Reviewer 2 Report
Malnutrition in patients with liver cirhosis is associated with poor prognosis bu is often underdiagnosed.
The submitted manuscript provider concise sumarycznie pathopysiology, diagnostics and therapeutic strategies of this significant clinical issue.
The paper is well written and includes 236 current references.
Author Response
We thank the reviewer for his/her favorable comments.
Malnutrition in patients with liver cirhosis is associated with poor prognosis bu is often underdiagnosed.
The submitted manuscript provider concise sumarycznie pathopysiology, diagnostics and therapeutic strategies of this significant clinical issue.
The paper is well written and includes 236 current references.
Reviewer 3 Report
To the authors
General:
This review is based on the existing literature which is extensively being cited. Comprehension needs to be improved. The data are described but need to be put into context and need to be weighted to arrive at “the current understanding of malnutrition in liver cirrhosis”. Only then this review will add substantially to the existing reviews.
The manuscript should be shortened by 25%. It needs to be read carefully by an independent expert to resolve some mistakes that are more than typos (e.g., L 120: brain chained amino acids; L 202: elevated; …).
Major:
Line 28/29: “malnutrition is a key feature and an important complication of the disease”
This is an interesting aspect and should be elaborated: complication “independently” being treated as a complication and key feature maybe treatable as part of treatment of the disease à how?
How does this emerge from table 1, which includes many studies. Can the table be broken up to make it digestible.
In order to arrive at a current understanding of malnutrition in cirrhosis (rather that lining up existing evidence) paragraphs 2.3, 2.5, 2.6 and 2.7 should be merged. Sarcopenia is mentioned in the introduction but significant advances in muscle metabolism in cirrhosis are not mentioned yet and should be included. Then maybe unresolved questions may arise.
The same holds true for the problem of obesity throughout the manuscript – either include and elaborate or leave out.
Section 2.8
Too general. RFH is the recommended screening tool. Then ESPEN or GLIM assessment can be performed with quite different results. Should another assessment tool be used, which? Is this an open question with a significant impact of the entire problem of key feature as compared to complication? Sarcopenia is too briefly discussed.
Line 208- 211: Common malnutrition screening tools for the general hospital population however do not perform well in cirrhosis. [172] Therefore, screening strategies need to identify specific patient populations, such as cirrhotic patients, in whom additional malnutrition screening is necessary. [172].
This is not specific and does not help: RFH is recommended for screening. Assessment is the problem and a gold standard for such an assessment is missing.
Section 3
The major evidence and the current recommendations are included. They should be weighed against each other and according to the different patient populations (the latter already being addressed).
Example: I guess it is too general to state “For cirrhotic patients with hepatic encephalopathy, vegetarian protein, which is rich in BCAA, is considered to be the ideal protein source. [194,232,233]“. First, all other reasons for HE need to be addressed – many of them are related to nutrition and then this statement may hold true for those that are intolerant to a regular high protein diet.
Minor:
Line 34-43: Quite general. Detection and gold standard may be more of a problem than missed diagnosis – especially if malnutrition is implicated as key feature of cirrhosis.
Line 61-63: This review summarizes the current knowledge on pathogenesis of malnutrition in cirrhosis and discusses the best strategies to diagnose malnutrition in order to raise awareness for this still often neglected complication of cirrhosis. phrase more careful: … often underappreciated aspect …
Line 78-87: In patients with liver cirrhosis, decreased energy and protein intake are the commonest reason leading to malnutrition. [47-53]
The paragraph list mechanisms upstream of decreased energy and protein intake --> they are the reasons
Line 88-91: first sentence --> grammar? “insufficient” --> “ineffective”?. Include ghrelin in figure 2. Ghrelin has posttranslational modifications --> include/discuss/refer
Table 2: Difficult to comprehend, because details of studies are missing. But it has too much detail to reference. Maybe adjust to format of table 1.
Line 104 – 106: Treating portal hypertension by the placement of a transjugular intrahepatic portosystemic shunt leads to improvement of fat free mass. [84-86]
That is only true for some TIPS-patients, which than have a much better prognosis. In this context Line 108: nutrient processing? Or rather protein-energy-metabolism and catabolism vs anabolism and microbiome.
Line 119 – 124: rephrase according to mechanism: decreased glycogen storage –> increase gluconeogenesis –> secondary protein breakdown … … -> fasting intolerance
Author Response
We thank the reviewer for his/her favorable and valuable comments. The responses to each point can be found below and in the revised manuscript file.
This review is based on the existing literature which is extensively being cited. Comprehension needs to be improved. The data are described but need to be put into context and need to be weighted to arrive at “the current understanding of malnutrition in liver cirrhosis”. Only then this review will add substantially to the existing reviews.
We carefully reassessed the text of the review to improve comprehension and to put data into context. The changes can be seen in the version including track changes.
The manuscript should be shortened by 25%.
We did our best to condense the text of the review and take out unnecessary passages and at the same time to incorporate the suggested additions by you and reviewer 1. We removed about 50 references but added another 20 according to the reviewers comments. We moved 3 tables to the supplementary file but we added one figure according to the suggestion of reviewer 1 and the graphical abstract according to the suggestion of the editor. Therefore, we have to apologize that the lengths of the manuscript is unchanged.
It needs to be read carefully by an independent expert to resolve some mistakes that are more than typos (e.g., L 120: brain chained amino acids; L 202: elevated; …).
We performed careful proofreading and hopefully found and corrected all typos.
Major:
Line 28/29: “malnutrition is a key feature and an important complication of the disease” This is an interesting aspect and should be elaborated: complication “independently” being treated as a complication and key feature maybe treatable as part of treatment of the disease à how?
Thank you for this comment. We reworded these sentences on the implication of this distinction in the introduction.
Please see lines 32-25 and 49-56 and we also take this up again in the diagnosis section (line 207) and in the summary (line 344)
How does this emerge from table 1, which includes many studies. Can the table be broken up to make it digestible.
We agree with your comment and decided that this table may not be ideally suited for the main text. In order to shorten the length review, we decided to streamline this table by excluding sarcopenia and osteodystrophy and to take this table out of the main manuscript and only display it as a supplementary table to improve readability of the review
In order to arrive at a current understanding of malnutrition in cirrhosis (rather that lining up existing evidence) paragraphs 2.3, 2.5, 2.6 and 2.7 should be merged. Sarcopenia is mentioned in the introduction but significant advances in muscle metabolism in cirrhosis are not mentioned yet and should be included. Then maybe unresolved questions may arise.
We took up this advice and restructured this section completely. Please kindly refer to the revised manuscript. We decided not to expand too much on sarcopenia, since the topic of the review is “malnutrition” and two excellent reviews were published only a few months ago on prevalence pathogenesis and therapy in Alimentary Pharmacology & Therapeutics and on diagnosis in Nutrients on the topic of sarcopenia in cirrhosis. We added these references to our review.
See line103-189
The same holds true for the problem of obesity throughout the manuscript – either include and elaborate or leave out.
We discussed this point in detail and in the light of your comment to considerably shorten the manuscript we decided to omit obesity and to state in the introduction that this is beyond the scope of this review.
Section 2.8
Too general. RFH is the recommended screening tool. Then ESPEN or GLIM assessment can be performed with quite different results. Should another assessment tool be used, which? Is this an open question with a significant impact of the entire problem of key feature as compared to complication? Sarcopenia is too briefly discussed.
Line 208- 211: Common malnutrition screening tools for the general hospital population however do not perform well in cirrhosis. [172] Therefore, screening strategies need to identify specific patient populations, such as cirrhotic patients, in whom additional malnutrition screening is necessary. [172].
This is not specific and does not help: RFH is recommended for screening. Assessment is the problem and a gold standard for such an assessment is missing.
We completely restructured the diagnosis section (also in accordance with the comments from reviewer 1) and added a new figure – a flowchart to make it more specific and useful for clinical practice. Please kindly refer to the revised manuscript.
Please see lines 207-240
Section 3
The major evidence and the current recommendations are included. They should be weighed against each other and according to the different patient populations (the latter already being addressed).
Example: I guess it is too general to state “For cirrhotic patients with hepatic encephalopathy, vegetarian protein, which is rich in BCAA, is considered to be the ideal protein source. [194,232,233]“. First, all other reasons for HE need to be addressed – many of them are related to nutrition and then this statement may hold true for those that are intolerant to a regular high protein diet.
We rewrote this section according to your suggestions to weight and interpret the information and to make it more specific.
Please see lines 244-342
Minor:
Line 34-43: Quite general. Detection and gold standard may be more of a problem than missed diagnosis – especially if malnutrition is implicated as key feature of cirrhosis.
The section of the introduction was reworded according to your suggestion to make it more specific.
Please see lines 40-65
Line 61-63: This review summarizes the current knowledge on pathogenesis of malnutrition in cirrhosis and discusses the best strategies to diagnose malnutrition in order to raise awareness for this still often neglected complication of cirrhosis. phrase more careful: … often underappreciated aspect …
We changed the wording according to your suggestion
Line 78-87: In patients with liver cirrhosis, decreased energy and protein intake are the commonest reason leading to malnutrition. [47-53]
The paragraph list mechanisms upstream of decreased energy and protein intake --> they are the reasons
We are not complete sure if we understand your comment correctly. The paragraph was reworded to make our point clearer. If we misunderstood you, please let us know and we are happy to change the text accordingly.
Please refer to line 74-77
Line 88-91: first sentence --> grammar? “insufficient” --> “ineffective”?. Include ghrelin in figure 2. Ghrelin has posttranslational modifications --> include/discuss/refer
We included Ghrelin in figure 2 and expanded on ghrelin. Since there was a review published only some months ago we decided to quote this review for further details.
Please refer to line 92-98 and 179-184
Table 2: Difficult to comprehend, because details of studies are missing. But it has too much detail to reference. Maybe adjust to format of table 1.
We reformatted the table according to your suggestion and adjusted it to table 1 and moved it to the supplements to reduce the length of the manuscript.
Line 104 – 106: Treating portal hypertension by the placement of a transjugular intrahepatic portosystemic shunt leads to improvement of fat free mass. [84-86]
That is only true for some TIPS-patients, which than have a much better prognosis. In this context Line 108: nutrient processing? Or rather protein-energy-metabolism and catabolism vs anabolism and microbiome.
The sentence was rephrased according to your suggestion.
Please see line 136-144
Line 119 – 124: rephrase according to mechanism: decreased glycogen storage –> increase gluconeogenesis –> secondary protein breakdown … … -> fasting intolerance
The statement was rephrased according to your suggestion.
Please see line 129-135
Round 2
Reviewer 3 Report
To the authors
My comments have been addressed.
Author Response
Thank you for reviewing our manuscript.
The numbers regarding the prevalence of malnutrition have been harmonized in the abstract and in the text and the missing supplementary table has been added.